# Parametrizing the mixing by clear air turbulence in the chemistry climate model EMAC and its respective radiative impact

Chun Hang Chau<sup>1</sup>, Peter Hoor<sup>1</sup>, Katharina Kaiser<sup>2</sup>, and Holger Tost<sup>1</sup>

Correspondence: Chun Hang Chau (cchau@uni-mainz.de)

Abstract. The Earth's radiation budget is found to be sensitive to changes in the upper troposphere and lower stratosphere (UTLS) chemical composition. Stratosphere-troposphere exchange is the major process that influences the UTLS chemical composition with remaining uncertainties in current climate-chemistry models. This exchange could be e.g., facilitated by clear air turbulence (CAT), as it leads to diabatic mixing of chemical tracers between stratosphere and troposphere. In this work, we examine the sensitivity of vertical mixing by CAT on the UTLS chemical composition and its corresponding radiative impact by implementing a newly developed submodel parametrizing turbulent mixing in the free troposphere and stratosphere within the climate chemistry model EMAC. This submodel parametrizes the vertical mixing by CAT based on a newly introduced turbulence diagnostic MoCATI. MoCATI shows a comparable performance with the well-established Ellrod-Knox index. Simulations are conducted with EMAC-QCTM to examine the sole impact of mixing, without taking the potential feedback into account. Results show that the radiatively active ozone in the UTLS is most sensitive to the vertical mixing of CAT and is significantly reduced by 10 to 20% by the CAT submodel. This modification is not a pure result of the physical mixing but also the chemical feedback of other modified tracers. The tracer mixing through CAT also changes the atmospheric chemistry by shortening the CH<sub>4</sub> lifetime and changing the O<sub>3</sub> becoming relatively sensitive to NO<sub>x</sub>. It also leads to potential surface radiative heating and radiative cooling at the top of the atmosphere. The global average radiative effect is about  $-0.2 \text{ W/m}^2$ .

#### 15 1 Introduction

The upper troposphere and lower stratosphere (UTLS) plays a crucial role on the Earth's radiation budget (Forster et al., 2021). Changes in the chemical composition in the UTLS could lead to changes via the radiation locally as well as at the surface. Ozone and water vapour near the tropopause are found to have more radiative influence on the surface compared to other well-mixed greenhouse gases (Riese et al., 2012; Lacis et al., 1990). Removal of ozone above the tropopause could lead to local cooling (Randel et al., 2007). The surface temperature is also highly sensitive to changing ozone concentration in the UTLS, increasing ozone in this region could increase the surface temperature (Forster and Shine, 1997; Lacis et al., 1990). Water vapour also has complex impacts on the Earth's radiation. Increasing stratospheric water vapour could lead to local cooling (Forster and Shine, 1999; Randel et al., 2007) and surface warming (Solomon et al., 2010).

<sup>&</sup>lt;sup>1</sup>Institute for Atmospheric Physics, Johannes Gutenberg University Mainz, Mainz, Germany

<sup>&</sup>lt;sup>2</sup>Aerosol Chemistry Department, Max Planck Institute for Chemistry, Mainz, Germany

50

Since stratosphere-troposphere exchange (STE) is one of the major processes that influence the UTLS chemical composition (Holton et al., 1995; Stohl et al., 2003), it is crucial to improve the representation of STE in the model. Previous studies show that the simulated upper tropospheric ozone budget depends on the model representation of STE (Stevenson et al., 2006). A key feature for STE is the occurrence of vertical shear, which has been recently identified constituting a layer of high probability of shear occurrence around the local tropopause (Kaluza et al., 2021). Vertical shear is a key mechanism to initiate turbulence and thus mixing. Turbulent mixing in the UTLS is one of the pathways of the STE (Holton et al., 1995) and it is not represented in several state-of-the-art chemistry climate models, such as EMAC (Jöckel et al., 2016). In ECHAM5, the base model of EMAC, the turbulence scheme was designed for the surface boundary layer only; it sets a constant asymptotic mixing length in the boundary layer and simply assumes it decreases exponentially with height, approaching 1 m in the lower stratosphere (Roeckner et al., 2003), resulting in a dampening effect on the turbulence in the UTLS, which normally does not mix tracers significantly. Therefore, this study presents a parametrization of tracer mixing by clear air turbulence (CAT), which is commonly found in the UTLS and could lead to rapid mixing of the chemical composition between the troposphere and stratosphere (Dutton and Panofsky, 1970; Esler and Polvani, 2004; Traub and Lelieveld, 2003), for EMAC.

CAT represents the turbulence in the free atmosphere that occurs in a cloud-free region (Ellrod et al., 2003). Kelvin-Helmholtz instability, which is a result of vertical wind shear (Kunkel et al., 2019), is the major mechanism that leads to CAT formation (Watkins and Browning, 1973; Ellrod and Knapp, 1992). Considering turbulence is not explicitly represented by global scale numerical models, CAT is usually forecast using diagnostics that are based on related larger-scale mechanisms. A previous study has shown that turbulence diagnostics are reliable for forecasting CAT operationally in NWP models (Sharman et al., 2006). However, each of the turbulence diagnostics has its own advantages and disadvantages. Ellrod index (TI) is one of the most commonly used CAT diagnostics in the aviation sector. It is based on the meso-scale condition that leads to CAT including vertical wind shear and deformation of the horizontal wind (Ellrod and Knapp, 1992). The Ellrod-Knox index is the improved version of the TI by introducing an additional divergence trend term (Ellrod and Knox, 2010). Williams and Storer (2022) confirmed that turbulence diagnostics are capable of diagnosing CAT and its response to climate change in climate models with coarser resolution. Another recent study from Chau et al. (2025) shows that the turbulence diagnostic calculated from the grid scale wind field matches well with a detailed sub-grid scale turbulence scheme in the UTLS for a regional forecast model.

CAT is expected to become more frequent and intense under climate change because of the more sheared atmosphere (Williams, 2017). The vertical wind shear in the North Atlantic region has increased by 15% between 1979 and 2017 (Lee et al., 2019). Several studies show that the strength of CAT will be stronger in the North Atlantic and East Asia under different climate change scenarios (Williams and Joshi, 2013; Smith et al., 2023; Hu et al., 2021). Therefore, there is a necessity to study how CAT could potentially change the atmosphere. Previous studies have examined the sensitivity of the vertical mixing in a Lagrangian approach, concluding that water vapour and ozone are most sensitive to vertical mixing and lead to a significant impact on the radiation budget (Riese et al., 2012). The main objective of this study is to examine the sensitivity of vertical turbulent mixing on the UTLS chemistry and radiation in an Eulerian approach. For this purpose, we introduce a new MESSy submodel to parametrize the CAT-induced vertical mixing of tracers in the UTLS. We demonstrate the possibility

https://doi.org/10.5194/egusphere-2025-5382 Preprint. Discussion started: 12 November 2025

© Author(s) 2025. CC BY 4.0 License.

of parametrizing CAT mixing based on turbulence indices. In order to examine the sole impact of CAT but not its feedback, the simulations are conducted in a QCTM mode (Deckert et al., 2011) to ensure identical model dynamics independent of the chemical composition.

This paper consists of three further sections after the introduction: Section 2 introduces the EMAC model configuration and the newly developed CAT submodel. Section 3 presents the results and discusses the CAT submodel in terms of redistributing chemical composition and radiative impact. Section 4 summarises the findings and draws conclusions with a future outlook.

#### 65 2 Method

## 2.1 EMAC Model Description

EMAC (ECHAM/MESSy Atmospheric Chemistry; Jöckel et al., 2006; Jöckel et al., 2010) is a state-of-the-art chemistry climate model that allows flexible model configuration via different submodels. It combines the general circulation model ECHAM5 (Roeckner et al., 2003) with the Modular Earth Submodel System (MESSy; Jöckel et al.,2005) which allows users to include different physical and chemical processes via a namelist interface. EMAC is also able to operate in quasi chemistry-transport model mode (QCTM; Deckert et al.,2011), which allows decoupling between chemistry and dynamics in the model, suppressing the feedback between chemistry and dynamics to better quantify the signal from a particular process. For our study, EMAC is operated in T42L90MA resolution (Giorgetta et al., 2006). It is a configuration designed for the middle atmosphere with a horizontal resolution of T42 (approximately 2.8°× 2.8°) and 90 vertical hybrid pressure levels up to 0.01 hPa.

#### 75 **2.2** Experimental setup

Two simulations: QCTM-MIX and QCTM-NOMIX are performed in this work. Two decadal simulations with gas-phase chemistry are performed using the EMAC-QCTM to examine the long-term impact of the CAT submodel without the dynamical feedback from chemistry. Both simulations are performed with the perpetual year of 2014 for 10 years, prescribing the emission, sea surface temperature and sea ice concentration of 2014 repetitively. The chemistry setup is adapted from the RC1-base-07 of Jöckel et al. (2016), which is the Ref-C1 of the CCMI (Eyring et al., 2013). The emissions are prescribed by the CCMI-2022 Ref-D1 (Plummer et al., 2021), which is a historical hindcast. The sea surface temperature and sea ice concentration are prescribed by the AMIP-II data set (Taylor et al., 2000). Both simulations are initialized with the ERA-Interim reanalysis data (Dee et al., 2011).

QCTM-NOMIX with the CAT submodel disabled is used as a reference simulation, while the QCTM-MIX enables the CAT submodel to examine the impact of the CAT tracer mixing. The details for the CAT submodel are discussed in Section 2.3.

## 2.2.1 Radiation (RAD) submodel of EMAC

In EMAC, the radiation scheme (RAD) is a re-implementation of the ECHAM5/ECHAM6 radiation codes with more flexibility (Dietmüller et al., 2016). It calculates the radiation depending on radiatively active tracers, including CO<sub>2</sub>, CH<sub>4</sub>, O<sub>3</sub>, N<sub>2</sub>O<sub>5</sub>,

110

CFC-11 and CFC-12. It also requires input parameters like water vapour, cloud cover, clear-sky index, cloud optical properties, aerosol optical properties, and orbital parameters provided by the model. RAD allows for both online and offline radiation calculations, i.e., using either prognostic variables (tracers) or external data sources for the radiatively active species. To run EMAC in the QCTM mode, climatological values or distributions of the radiatively active gases are utilised for the radiation calculation to decouple the feedback between the gases and the radiation scheme. Water vapour also needs to be decoupled from the chemistry to prevent inconsistency in model dynamics. Therefore, the mixing scheme of the new submodel CAT (Section 2.3) does not include water vapour at this stage. RAD also provides an option to calculate the radiative disturbances from different sets of radiatively active gases by calling the radiation routine multiple times within one model time step. The temperature feedback will be provided by the first call. This option provides an opportunity to examine the pure radiative impact of the CAT mixing without dynamical feedback within EMAC-QCTM. In both simulations, a total of six calls are set. For the first call, which provides the temperature feedback of the model, prescribed values for the radiatively active gases are used. For the second call, the radiation calculation uses all the online interactive chemistry output provided by EMAC. To isolate the individual contribution of each gas, four additional radiation calls are set. In each call, only the specified gas uses the online output from the model, the other gases remain unchanged using the prescribed values. Detailed information about the applied values and the dataset is given in Table 1. The prescribed values for CO<sub>2</sub>, CH<sub>4</sub>, N<sub>2</sub>O, CFC-11 and CFC-12 are provided by the Global Monitoring Laboratory of NOAA. The prescribed O<sub>3</sub> is taken from the climatology of Paul et al. (1998).

**Table 1.** Summary of the performed radiation calls and the applied values [mol/mol] for the active tracers

| Call no.                | $CO_2$      | $O_3$               | $\mathrm{CH}_4$ | $N_2O$      | CFC-11      | CFC-12      |
|-------------------------|-------------|---------------------|-----------------|-------------|-------------|-------------|
| RAD01                   | 397.34e-6   | (Paul et al., 1998) | 1.82254e-6      | 327.09e-9   | 230e-12     | 520e-12     |
| RAD02                   | interactive | interactive         | interactive     | interactive | interactive | interactive |
| RAD03 (O <sub>3</sub> ) | 397.34e-6   | interactive         | 1.82254e-6      | 327.09e-9   | 230e-12     | 520e-12     |
| RAD04 ( $CO_2$ )        | interactive | (Paul et al., 1998) | 1.82254e-6      | 327.09e-9   | 230e-12     | 520e-12     |
| RAD05 ( $CH_4$ )        | 397.34e-6   | (Paul et al., 1998) | interactive     | 327.09e-9   | 230e-12     | 520e-12     |
| RAD06 ( $N_2O$ )        | 397.34e-6   | (Paul et al., 1998) | 1.82254e-6      | interactive | 230e-12     | 520e-12     |

# 105 2.3 New MESSy submodel CAT for clear air turbulence mixing

This section introduces the newly developed submodel CAT of MESSy. The new CAT submodel is developed to parametrize the vertical mixing of tracers caused by clear air turbulence in the UTLS. It allows vertical mixing of tracers based on a 2-layer mixing algorithm between 500 hPa to 70 hPa, considering that CAT occurs most frequently in the UTLS (Dutton and Panofsky, 1970). The mixing scheme of CAT uses a turbulence diagnostic (CAT Index) to serve as a mixing coefficient and provides two options for now: the Ellrod-Knox index (Ellrod and Knox, 2010) and a newly introduced Modified CAT Index (MoCATI). The Ellrod-Knox index is a widely tested and used turbulence diagnostic, which is based on the grid scale wind field data including

125

deformation (DEF), vertical wind shear (VWS), and divergence trend (DVT):

Ellrod-Knox index = 
$$\underbrace{\left[\left(\frac{\partial u}{\partial x} - \frac{\partial v}{\partial y}\right)^{2} + \left(\frac{\partial v}{\partial x} + \frac{\partial u}{\partial y}\right)^{2}\right]^{\frac{1}{2}}}_{\text{DEF}} \cdot \underbrace{\left(\left|\frac{\partial u}{\partial z}\right|^{2} + \left|\frac{\partial v}{\partial z}\right|^{2}\right)^{\frac{1}{2}}}_{\text{VWS}} + \underbrace{C\left[\left(\frac{\partial u}{\partial x} + \frac{\partial v}{\partial y}\right)_{h2} - \left(\frac{\partial u}{\partial x} + \frac{\partial v}{\partial y}\right)_{h1}\right]}_{\text{DEF}} \quad (1)$$

The subscripts h1 and h2 represent the selected time interval for the divergence trend. C is a weighting constant to prevent the divergence trend term from dominating the whole Ellord-Knox index.

MoCATI is developed on top of the Ellrod-Knox index. It is a modification of the Ellrod-Knox index including static stability:

120 MoCATI = 
$$\frac{N_{lim}^2 - N^2}{N_{lim}^2}$$
 · Ellrod-Knox index (2)

The  $N^2$  is the Brunt-Väisälä frequency and  $N^2_{lim}$  is a limitation threshold to modify the whole stability term, considering that a stable environment will suppress the formation of turbulence. In addition, the mixing will be switched off above the  $N^2_{lim}$ .

**Figure 1.** Schematic of the tracer mixing algorithm of the CAT submodel. F represents the flux of mixing and n represents the model vertical level.

A schematic of the tracer mixing algorithm is shown in Figure 1. The flux of the mixing between level n and level n+1 depends on the mixing ratio  $(\chi)$  and the CAT index at the boundary of both layers. The flux from level n to n+1 can be express

$$F_{n \to n+1} = \chi \cdot \text{CAT Index}_{boundary} \tag{3}$$

The CAT index at the boundary can be calculated by:

$$\text{CAT Index}_{boundary} = \frac{\text{CAT Index}_n + \text{CAT Index}_{n+1}}{2} \cdot \frac{\Delta t}{t_{norm}} \tag{4}$$

The  $\Delta t$  is the time step length and the  $t_{norm}$  is a time normalization factor. The whole term is used to moderate the strength of mixing.

The CAT submodel allows changes in different parameters via a simple namelist, for which an example and the detailed explanation of the respective parameters can be found in the electronic supplement.

#### 3 Results and Discussion

## 3.1 Evaluation of the newly introduced MoCATI

In order to examine the validity of the MoCATI, we performed statistical tests and calculated the relative frequency distribution of the turbulence diagnostics following the approch of Kaluza et al. (2022), using the ERA5 data for three boreal winters (DJF) from December 2016 to February 2019 over the North Atlantic to calculate and compare the distribution between the wellestablished Ellrod-Knox index and the MoCATI, but instead of just covering the flight tracks as in Kaluza et al. (2022), we perform calculations on the whole North Atlantic. The North Atlantic winter is found to have the most vertical wind shear (Kaluza et al., 2021). Figure 2 shows a map of the domain from 60° W to 0° W and 35° N to 60° N for the Ellrod-Knox index and MoCATI at the lapse rate tropopause on 01 December 2016. Both indices show a similar distribution, although the magnitude of MoCATI is weaker considering it is constrained by the additional stability term. Figure 3 shows the Taylor diagram using the Ellrod-Knox index as a reference. It shows the correlation coefficient, normalized standard deviation and the normalized root-mean squared difference (both normalized with the reference standard deviation from the Ellrod-Knox index). As expected, the MoCATI shows a similar behaviour on these statistical metrics since it is a modification of the Ellrod-Knox index. The MoCATI is highly correlated, with a similar normalized standard deviation close to the perfect 1.0 line, and a small RMSD of half of the reference standard deviation. Figure 3 also includes the Ellrod index (TI), vertical wind shear (VWS) and deformation (DEF). Although the TI, VWS and DEF are included as part of the Ellrod-Knox index, since an additional divergence trend term is added, they show a lower correlation with the Ellrod-Knox index compared to the MoCATI. Figure 4 shows the relative frequency distribution of the ln(MoCATI) and ln(Ellrod-knox index) by geometric height and distance relative to the lapse rate tropopause. Both indices show a similar distribution, with a rightward shift (i.e., stronger turbulence) at lower altitudes and a leftward shift at higher altitudes. MoCATI has a more prominent leftward shift at higher altitudes considering the additional static stability term which constrains the index. We also perform a ROC (Receiver operating characteristic) analysis (Fawcett, 2006) to examine the performance of the Ellrod-Knox index and MoCATI using TI as the classification threshold (Sharman et al., 2006). Both indices show a similar performance under different thresholds (Figure 5). To conclude, the newly introduced MoCATI shows a comparable performance with the well-established Ellrod-Knox index, but takes the static stability into account, modifying the effective mixing under specific temperature profile conditions.

Figure 2. Map of the Ellrod-Knox index and MoCaTI at the lapse rate tropopause on 01 Dec 2016 1600 UTC.

**Figure 3.** Taylor diagram showing the correlation coefficient, normalized standard deviation and normalized root mean square difference (RMSD) using the Ellrod-Knox index as reference.

**Figure 4.** Relative frequency distribution for MoCATI (left) and Ellrod-Knox index (right) in geometric height (top) and relative to the lapse rate tropopause (bottom). Dashed lines indicate the 95th percentile (black) and 90th percentile (grey).

**Figure 5.** ROC curves showing the probability of false detection and true detection for Ellrod-Knox index and MoCATI using different TI as threshold: T1 = null, T2 = light, T3 = moderate, T4 = severe and T5 = extreme. The brackets represent the percentage of data that reaches the corresponding threshold. AUC represents the area under the curve.

## 3.2 Simulation results

## 3.2.1 Comparison of simulated O<sub>3</sub> with SWOOSH observations

**Figure 6.** Cross section of the annual ozone zonal mean difference for (a) QCTM-NOMIX minus SWOOSH, (b) QCTM-MIX minus SWOOSH, and (c) QCTM-NOMIX minus QCTM-MIX. The contour lines represent the mixing ratio of O<sub>3</sub> in ppmv of the respective QCTM simulation.

Figure 6a and b show the annual mean difference of ozone between both simulations and the satellite-based database SWOOSH (Davis et al., 2016). By applying the CAT submodel, the overestimated ozone over the polar regions in the stratosphere is significantly reduced. Figure 6c shows the annual mean difference between both simulations. The vertical mixing by CAT in QCTM-MIX reduces the ozone significantly between 50 to 100 hPa compared to QCTM-NOMIX, the details are discussed further in Section 3.2.2. A previous study from Righi et al. (2015) shows that EMAC underestimates the ozone hole in Antarctica, causing a warm bias in the southern hemispheric stratosphere. QCTM-MIX shows a better consistency with SWOOSH in this region for ozone. As will be described later, Fig. 12 shows the annual mean difference in the net radiation between both simulations. The local cooling caused by the changes in ozone in this region might partly resolve the warm bias in this region. In order to clarify whether CAT could improve the representation of ozone in EMAC, a detailed investigation is needed. However, it is beyond the scope of this study.

## 170 3.2.2 Difference in tracer distributions

**Figure 7.** Annual zonal mean O<sub>3</sub>, CH<sub>4</sub>, CO<sub>2</sub> and N<sub>2</sub>O profile of QCTM-NOMIX (first column), absolute difference between both simulations (centre column), and relative percentage difference (right column). The black line indicates the tropopause.

185

Figure 7 shows the annual zonal mean mixing ratio and difference of O<sub>3</sub>, CH<sub>4</sub>, CO<sub>2</sub> and N<sub>2</sub>O between QCTM-MIX and QCTM-NOMIX to assess the sensitivity of tracers to the mixing algorithm of CAT. These major radiatively active gases were selected for the analysis to illustrate the different responses to CAT between spatially variable gases and well-mixed gases. As a result of enabling CAT, the O<sub>3</sub> in the UTLS is significantly reduced, especially in the vicinity of the tropopause with a 10 to 20% decrease (except below the tropical tropopause). The changes between 100 to 50 hPa at the extratropics could also be attributed to changes of the gradient by the enhanced vertical mixing at the tropics which leads to enhanced horizontal advection through the shallow branch of the Brewer-Dobson circulation. CAT also changed the O<sub>3</sub> by changing the chemistry (Figure 8a & b) in the upper stratosphere (above 50 hPa), causing up to 3% increase, and transport downward across the tropopause to the troposphere with maximum of 9% increase in the tropical upper troposphere and southern hemisphere. In terms of the mixing ratio, the largest absolute difference is located at around 75 hPa in the extratropical region, with up to 320 ppbv reduction of O<sub>3</sub>. For CH<sub>4</sub>, CAT leads to a significant increase in the UTLS up to 2.5% or 40 ppbv and a slight loss in the troposphere. The two gases exhibit an opposite behaviour with O<sub>3</sub> decreasing and CH<sub>4</sub> increasing in the UTLS since the O<sub>3</sub> mixing ratio is higher in the UTLS than in the troposphere whereas the CH<sub>4</sub> mixing ratio is higher in the troposphere. O<sub>3</sub> also has a more significant change compared to CH4 due to the steeper gradient. The changes in both gases are stronger in the southern hemisphere, potentially due to the stronger jet streams and fewer mountains, leading to a stronger vertical wind gradient. For CO<sub>2</sub>, since it is well-mixed with a long atmospheric lifetime, the effect of mixing is comparatively weak, given the resulting low vertical gradient. In contrast to this, the effect of CAT mixing on the N<sub>2</sub>O mixing ratio is slightly stronger, similar to CH<sub>4</sub>, but with a more homogeneous effect on both hemispheres.

Figure 8. Annual zonal mean difference (MIX - NOMIX) of  $O_3$  chemical production and chemical loss: (a) total production, (b) total loss, (c) loss by bromine, (d) loss by chlorine, (e) loss by hydrogen species, (f) loss by nitrogen species, (g) loss by atomic oxygen and (h) loss by peroxy radicals. The black line indicates the tropopause.

200

To examine whether the differences in the tracer distribution are attributed solely to the mixing of CAT or also influenced by the chemical feedback of other mixed tracers, we analyzed the chemical production and loss terms of ozone as well. The chemistry submodel MECCA (Sander et al., 2011) used in EMAC provided diagnostic output for some of the chemical species. Figure 8 shows the difference (MIX minus NOMIX) in ozone production, total loss and loss by species. With CAT mixing, the production of ozone (Figure 8a) is reduced and the loss is enhanced (Figure 8b) at the tropical tropopause, whereas the extra-tropical lower stratosphere (LS) shows an inverse behaviour. These results indicate that the reduction of ozone (Figure 7) at the tropical tropopause is enhanced by the chemical feedback, whereas the physical mixing is offset by the chemical feedback in the extra-tropical LS.

The chemical loss of ozone (Figure 8b) is enhanced below the tropical tropopause after including the CAT mixing parametrization. It is mainly contributed by  $HO_x$  (Figure 8e) and  $O(^1D)$  (Figure 8g). Figure 8g shows the difference in the chemical loss by atomic oxygen. This shows the loss of ozone by removing  $O(^1D)$  from the odd oxygen: ozone is photolysed to form  $O(^1D)$  via Equation R1,  $O(^1D)$  is then consumed with  $H_2O$  to form OH (R2). This also partly explains the increase of OH in Figure S2c in the supplement.

$$O_3 \xrightarrow{h\nu} O(^1D) + O_2$$
 (R1)

$$O(^{1}D) + H_{2}O \longrightarrow 2OH$$
 (R2)

Figure 8e shows the changes in the chemical loss by  $HO_x$ . This illustrates the increasing loss of ozone at the tropical tropopause due to the  $OH/HO_x$  catalytic cycle. Ozone is destroyed by Equations R3 and R4 with  $OH/HO_x$  as a catalyst. This cycle is enhanced due to the increasing  $HO_x$  concentration (Figure S2f in the supplement).

$$OH + O_3 \longrightarrow HO_2 + 2O_2 \tag{R3}$$

$$HO_2 + O_3 \longrightarrow OH + O_2$$

$$\overline{\text{Net: } 2O_3 \longrightarrow 2O_2}$$
(R4)

Mixing by CAT also reduces the chemical loss of ozone (Figure 8b) at the extra-tropical LS ( $\sim$ 100 to 50 hPa). Around half of the reduction is due to the reduction of HO<sub>x</sub> (Figure S2f) and NO<sub>x</sub> (Figure S2l and S2o) in the corresponding region. The reduction of HO<sub>x</sub> in the extra-tropics LS and NO<sub>x</sub> reduction in the UTLS decelerate the HO<sub>x</sub> cycle and NO<sub>x</sub>-related cycle. Halogens are responsible for the remaining reduction, especially in the Southern Hemisphere. The vertical mixing evened out the high concentrations at the poles, decelerating the catalytic cycle.

In addition, the change of the chemical regime of ozone is analyzed using the HCHO/NO<sub>2</sub> ratio (Formaldehyde to NO<sub>2</sub> ratio; FNR) and a novel diagnostic  $\alpha_{\text{CH}_3\text{O}_2}$  developed by Nussbaumer et al. (2022). FNR is a commonly used indicator that develops from the principle that formation of tropospheric ozone requires volatile organic compounds (VOCs) and NO<sub>x</sub> as its precursors (Acdan et al., 2023; Jin et al., 2020; Jiang et al., 2025). HCHO and NO<sub>2</sub> serve as the proxies of the VOCs and NO<sub>x</sub> respectively, considering their similar chemical lifetime could better show competition between species (Tonnesen and Dennis, 2000). The lower FNR indicates VOC sensitivity, while the higher FNR indicates NO<sub>x</sub> sensitivity. The  $\alpha_{\text{CH}_3\text{O}_2}$  represents the fraction of CH<sub>3</sub>O<sub>2</sub> forming HCHO via NO and OH relative to the competing pathway that forms CH<sub>3</sub>OOH via HO<sub>2</sub> which could be used to

identify the HCHO production pathways. It could also be used to study ozone sensitivity (Nussbaumer et al., 2024). It is based on the concept that the formation of HCHO from  $CH_3O_2$  with NO leads to  $O_3$  production, while the formation of  $CH_3OOH$  from  $CH_3O_2$  with  $CH_3OOH$  with CHOOH which generally only plays a minor role, will not contribute to  $CH_3OOH$  formation and therefore will be neglected when studying ozone sensitivity. It can therefore be calculated as shown in Equation. 5:

$$\alpha_{\text{CH}_3\text{O}_2} = \frac{k_{\text{CH}_3\text{O}_2 + \text{NO}} \times [\text{NO}]}{k_{\text{CH}_3\text{O}_2 + \text{NO}} \times [\text{NO}] + k_{\text{CH}_3\text{O}_2 + \text{HO}_2} \times [\text{HO}_2]}$$
(5)

Nussbaumer et al. (2023) shows that both the FNR and  $\alpha_{CH3O2}$  are good indicators for the ozone chemical regime in the upper troposphere. Figure 9 shows the 10-year climatology of the  $\alpha_{CH3O2}$ . CAT leads to slight changes in  $\alpha_{CH3O2}$  in different altitudes. The difference of  $\alpha_{CH3O2}$  (Figure 9a) shows a bi-modal distribution at the upper troposphere ( $\sim$ 200 hPa) and the lower troposphere ( $\sim$ 800 hPa). The Southern hemisphere exhibits a larger decrease with a maximum of 3%. The difference even reaches up to 6% during the JJA season in the southern polar region (not shown). The decreasing  $\alpha_{CH3O2}$  indicates the decrease of NO and increase of HO<sub>2</sub>, which is consistent with Figure S2l and S2f and the O<sub>3</sub> is relatively more sensitive to NO<sub>x</sub> (less sensitive to VOCs) than without the CAT mixing. Figure 10 shows the 10-year climatology for the FNR. The Southern hemisphere depicts a more substantial absolute increase in FNR in the lower troposphere( $\sim$ 800 hPa), while the upper troposphere ( $\sim$ 200 hPa) shows a percentage increase up to 26%. The changes above 200 hPa are neglected since the stratospheric ozone chemistry is significantly different than the troposphere. It shows consistent results with the  $\alpha_{CH3O2}$ , the increasing FNR indicates the O<sub>3</sub> chemical regime is becoming relatively more NO<sub>x</sub> sensitive (less VOCs sensitive). Both the FNR and  $\alpha_{CH3O2}$  show that the CAT mixing in the UTLS also affects the tropospheric chemistry.

Figure 9. Vertical profile of the 10-year climatology of  $\alpha_{CH_3O_2}$  in different regions: (a) absolute difference (MIX - NOMIX), (b) percentage difference, (c) profiles. The solid lines indicates profiles of MIX, the dashed lines indicate profiles of NOMIX. The different colors denote respective latitude bands.

**Figure 10.** Vertical profile of the 10-year climatology of the HCHO/NO<sub>2</sub> ratio (FNR) in different regions: (a) absolute difference (MIX - NOMIX), (b) percentage difference, (c) profiles. The solid lines indicate profiles of MIX, the dashed lines indicate profiles of NOMIX. The different colors denote respective latitude bands.

**Figure 11.** Vertical profile of the 10-year climatology of CH<sub>4</sub> lifetime (year) in different regions: (a) absolute difference (MIX - NOMIX), (b) percentage difference, (c) profiles. The solid lines indicate profiles of MIX, the dashed lines indicate profiles of NOMIX. The different colors denote respective latitude bands.

Furthermore, the chemical lifetime of CH<sub>4</sub> has been analyzed as well. Figure 11 shows the climatology of the methane lifetime. CAT mixing leads to a significant reduction of the CH<sub>4</sub> lifetime in the UTLS region, the tropics experienced a 15-year

( $\sim$ 8%) decrease. The global atmospheric lifetime of CH<sub>4</sub> also shortened from 7.31 years to 7.24 years. It is mainly due to the change in OH distribution by CAT. Figure S2c shows that the OH increases mainly in the lower stratosphere with 10% of increase at the tropical tropopause which coincides with the lifetime profile in Figure 11. Considering the extremely short atmospheric lifetime of OH, the changes are unlikely to be a direct result of the CAT mixing on OH. It must be a result of either the changes in the OH precursors or sinks. One of the possible pathways is that the increasing OH by O( $^{1}$ D) leads to a more efficient CH<sub>4</sub> oxidation, which produces more HCHO and HO<sub>2</sub> (based on the lower  $\alpha_{\text{CH}_{3}\text{O}_{2}}$ ). Photolysis of HCHO and HO<sub>2</sub> recycling then generates OH, establishing a HO<sub>x</sub>-CH<sub>4</sub>-HCHO feedback cycle.

To conclude, the CAT mixing clearly modifies the chemical regime of the  $O_3$  budget, shifting the  $O_3$  to become relatively sensitive to  $NO_x$ , as well as reducing the atmospheric lifetime of CH<sub>4</sub> by increasing OH.

## 3.2.3 Impact on radiation

The simulation results show that the CAT mixing of tracers leads to a radiative cooling on the global mean radiation fluxes at the top-of-the-atmosphere (TOA) of about 208.9 mW/m<sup>2</sup>. It is mainly induced by the change in ozone (-210.5 mW/m<sup>2</sup>). The other major greenhouse gases like CO<sub>2</sub>, CH<sub>4</sub> and N<sub>2</sub>O have a comparably small impact which either leads to a radiative cooling or heating effect at the TOA. The changes of the annual mean net radiation fluxes of different radiation calls are shown in Figure 12 and Figure 13. Each radiation call represents a different greenhouse gas scenario; a summary of the radiation calls is given in Table 1. Figure 12 illustrates the annual zonal mean difference of the net radiation fluxes. In general, CAT mixing leads to cooling above the tropopause and heating in the troposphere (Figure 12a). It is mainly contributed by ozone (Figure 12b), with almost identical behaviour as in Figure 12a. The loss of ozone in the UTLS shown in Figure 7a, leads to local and TOA radiative cooling as well as heating in the troposphere. CO<sub>2</sub> and CH<sub>4</sub> show mainly an opposite response compared to ozone, while CO<sub>2</sub> has a stronger radiative heating in the northern hemisphere, potentially because of anthropogenic activities. N<sub>2</sub>O shows a heating effect in almost all altitudes, except for the tropical lower troposphere. Figure 13 shows the annual mean difference of the net radiation fluxes at the TOA and surface. CAT leads to radiative cooling at the TOA and heating at the surface, mainly contributed by ozone. It is more pronounced along the storm tracks, where strong jet stream and associated wind shear favour turbulent mixing in the UTLS. At the TOA, CO<sub>2</sub> leads to cooling in the Southern Hemisphere and heating in the Northern Hemisphere, while the surface shows an inverse behaviour. CH<sub>4</sub> warms the tropics at the TOA, but cools at the surface. Changes in N<sub>2</sub>O result in heating at the TOA and the surface, except for the tropics.

Overall, CAT mixing leads to cooling in the Earth's radiation budget, mainly contributed by  $O_3$ , which is a greenhouse gas with spatial variability and also strongly active in the shortwave spectrum. In contrast, well-mixed greenhouse gases like  $CO_2$ ,  $CH_4$  and  $N_2O$  are rather insensitive to CAT mixing, with global mean TOA radiative effects of -0.2 mW/m<sup>2</sup>, 0.13 mW/m<sup>2</sup> and 1.62 mW/m<sup>2</sup> respectively.

We examined the impact of CAT on tracers and their associated radiative effects. Overall, CAT leads to an increase in outgoing LW radiation at the TOA, corresponding to a cooling effect. It is mainly driven by ozone. The other major greenhouse gases, such as CO<sub>2</sub>, CH<sub>4</sub>, and N<sub>2</sub>O which lead to changes in radiation of TOA are comparatively insensitive to CAT considering

they are well-mixed in the atmosphere. A previous study by Riese et al. (2012) shows similar results on the sensitivity of the greenhouse gases by mixing, where water vapour and ozone are notably more sensitive than CH<sub>4</sub> and N<sub>2</sub>O. However, due to differences in the mixing parametrization, it resulted in an opposite radiative impact. That study uses the Chemical Lagrangian Model of the Stratosphere (CLaMS) with a mixing algorithm based on atmospheric deformation (McKenna et al., 2002a, b; Konopka et al., 2004). This mixing scheme merges air parcels when they fall below the minimum separation distance. This deformation-induced mixing is a good approximation for stirring and quasi-isentropic mixing in the stably stratified stratosphere. Some more recent studies (Pommrich et al., 2014; Konopka et al., 2019) show that in the previous version of CLaMS which Riese et al. (2012) were using, the vertical transport within the troposphere is underestimated. This causes the mixing by CLaMS to be mainly attributed to quasi-horizontal isentropic transport which dominates between tropical UT and extra-tropical LS and within the stratosphere, which only reflects part of the potential mixing in the UTLS. The new CAT submodel, on the other hand, parametrizes the mixing by turbulence, a diabatic process that allows exchange across the tropopause. It includes not only the deformation but also vertical wind shear, divergence, and static stability. Also, the CLaMS chemistry is relatively simple compared to EMAC, as well as in describing small-scale dynamics. The mixing scheme of CAT is in turn, more sensitive to vertical mixing, while CLaMS is more sensitive to the isentropic (quasi-horizontal) mixing.

Figure 12. Annual zonal mean difference of net radiation fluxes for different radiation calls between two simulations: (a) RAD02, (b) RAD03 ( $O_3$ ), (c) RAD04 ( $O_2$ ), (d) RAD05 ( $O_3$ ), and (e) RAD06 ( $O_2$ ). The numerical values given are differences of the global mean TOA radiation. The black line indicates the tropopause

**Figure 13.** Annual mean difference of net radiation fluxes [mW/m<sup>2</sup>] at the TOA for (a) RAD02, (c) RAD03, (e) RAD04, (g) RAD05, and (i) RAD06; and at the surface for (b) RAD02, (d) RAD03, (f) RAD04, (h) RAD05, and (j) RAD06.

#### 4 Conclusion and Outlook

This study presents a new submodel for parametrizing vertical tracer mixing in the UTLS by Clear Air Turbulence (CAT), developed for the climate chemistry model EMAC, considering the dampening characteristics of the ECHAM5 turbulence scheme above the planetary boundary layer. The CAT mixing scheme is based on a two-layer mixing algorithm and the newly developed turbulence diagnostic MoCATI, which is a modification of the well-known Ellrod-Knox index by introducing static stability into the calculation. We conducted statistical tests and the relative frequency distribution of MoCATI and Ellrod-Knox index. The new index shows a comparable performance as the Ellrod-Knox index.

We performed two simulations (MIX and NOMIX) using the EMAC model in a QCTM configuration to examine the impact of CAT mixing parametrization on tracer distribution and its corresponding radiative impact. CAT leads to significant impacts on the non well-mixed gases like ozone in the UTLS. The changes are contributed by both the physical mixing and the chemical feedback from other tracers. ceO3 is significantly reduced in the UTLS and increases in the free troposphere as well as in the mid-stratosphere. The reduction in the tropics is a result of both the physical mixing and chemical feedback from other mixed tracers like odd oxygen and  $HO_x$  while the extra-tropics physical mixing is offset by the chemical feedback not only from odd oxygen and  $HO_x$ , but also from halogen species. Other major greenhouse gases including  $CO_2$ ,  $CH_4$  and  $N_2O$  are relatively insensitive to CAT considering their well-mixed characteristics and weak gradients particularly in the UTLS. It also shows that the CAT in the UTLS can change the chemical regime of  $O_3$  not only in the UTLS region, but also in the lower troposphere, by shifting to a relatively more  $NO_x$  sensitive (less VOCs sensitive) environment. Methane lifetimes are also found to be shortened from 7.31 years to 7.24 years (especially at the stratosphere with a 15-year difference) because of the more effective  $HO_x$  catalytic cycle. CAT mixing also reduces the ozone bias at the polar mid-stratosphere by comparing the results with SWOOSH.

Simulations show that CAT could lead to surface radiative heating and radiative cooling at the TOA. The TOA is expected to be 0.2W/m<sup>2</sup> cooler. It is mostly contributed by ozone since water vapour is not taken into account in the QCTM mode. Other major greenhouse gases show negligible impacts on the radiation budget. Considering the strengthening trend of CAT under climate change, the cooling effect of CAT mixing on tracers could potentially partially offset the warmer climate.

Even though water vapour is not taken into account in the current version of CAT to obtain consistent dynamics in the QCTM mode, it is planned to also consider the potential radiative impact of water vapour by extending the CAT submodel and include the mixing of water vapour and temperature. The water vapour is expected to increase in the UTLS considering its gradient. The CAT submodel is also expected to be extended to include and combine more turbulence diagnostics in the future, considering the pros and cons of each diagnostic.

Code availability. The model code of the EMAC climate chemistry model can be obtained by becoming a member of the MESSy consortium as described on the corresponding webpage https://messy-interface.org/.

Author contributions. CHC and HT conceptualized the study. HT and KK developed the CAT submodel with modifications from CHC. CHC and KK contributed to the submodel evaluation. CHC performed the simulations and analyzed the model results. CHC drafted the manuscript. CHC, HT, and PH discussed the results. All co-authors contributed to the review and final editing of the manuscript.

Competing interests. The contact author has declared that none of the authors has any competing interests

325

Acknowledgements. This work has been funded by the Deutsche Forschungsgemeinschaft (DFG, German Research Foundation) – TRR 301 – Project-ID 428312742 (project B01). The simulations were conducted using the supercomputer MOGON-NHR of Johannes Gutenberg University Mainz (https://hpc.uni-mainz.de/, last access: 15 September 2025).

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
