# Peer review of "Parametrizing the mixing by clear air turbulence in the chemistry climate model EMAC and its respective radiative impact"

_EGUsphere, 2025_

## Referee Comment (RC1)

Comments on "Parametrizing the mixing by clear air turbulence in the chemistry climate model EMAC and its respective radiative impact"

The manuscript "Parametrizing the mixing by clear air turbulence in the chemistry climate model EMAC and its respective radiative impact" by Chau et al. submitted to the EGUsphere describes the simulation results of a chemistry climate model (at somewhat coarse resolution) with and without a parameterization for clear-air turbulence (CAT)-related mixing, with a recently developed diagnostic index that accounts the effect of stratification on the turbulence intensity. The simulations are run without the feedback of chemistry to the dynamics. The effects of the new CAT parameterization is examined with respect to ozone, methane, CO2 and NOx. Some model improvements on the ozone concentration is demonstrated when compared to satellite-based global dataset. For other chemical species, the results are mainly based on comparing simulation differences, without observational benchmark.

Overall, I think the scientific novelty is somewhat lacking, but I do think it is important to document the performance of the new CAT-mixing parameterization. So if this kind of model evaluation work falls within the scope of EGUsphere, I would recommend major revisions. I must also confess that I am no expert on atmospheric chemistry, so my comments are mainly based on the CAT parameterization.

**Major comments**

- 1. Please describe your CAT-mixing submodel in more details. This manuscript is about evaluating this specific parameterization, so do elaborate on the details, and not just place it in the supplementary material. For example, does the CAT mixing submodel act on all vertical elevations of the atmosphere, does it include the boundary layer? What values are set for the parameters (for example h1 and h2 in Eq. 1 and t\_norm in Eq. 4), and based on what principles are these values chosen?
- Eq. 3, the vertical flux F ≡ w'χ' should be a vertical velocity scale times the unit of the tracer, which means that the CAT Index should have units of [m/s], right? But based on Eq. 4, the CAT Index as the same unit as CAT defined in Eqs. 1 and 2, which is [1/s]. Correct me if I am wrong, but what is with the dimensional inconsistency?
- 3. The MoCATI adopt a correction based on the Vaisala frequency alone (i.e.,  $1-N^2/N_{lim}^2$ ). First of all, please define the Brunt-Vaisala frequency, did you use the dry Brunt-Vaisala

- frequency or the moist one? Secondly, I am curious why the stratification correction, wouldn't a Richardson number-based correction be better?
- 4. Section 3.2, before you present the results for the chemistry species, I think it would help to present a latitude-height distribution of CAT mixing, i.e., the CAT Index\_{boundary} in your Eq. 3, so that the readers have a sense of where CAT-induced mixing occur. This would also assist the interpretation of the following figures.

**Minor comments**

- 1. Abstract, please do not abbreviate at first appearance, such as MoCATI, EMAC-QCTM.
- 2. Line 12, what "modified tracers", modified by what?
- 3. Line 13, not sure what you mean by "changing the O3 becoming relatively sensitive to NOx".
- 4. Lines 27-28, "vertical shear, which has been recently identified constituting a layer of high probability of shear", please improve your grammar.
- 5. Line 36, remove "for EMAC", you are talking about a physical process, right? Not just for a numerical model.
- 6. Lines 42 and 45, I hope both definitions are given below.
- 7. Line 51, at which elevation? Please be more specific.
- 8. Line 58, what is "MESSy"?
- 9. Line 60, again, what is "QCTM"?
- 10. Line 116, so what time interval was selected for this study, and why?
- 11. Line 129, How did you set t\_norm? It is also strange that the turbulent flux (tracer transported per area per time) should depend on the model time step. Some explanation is needed before you proceed with testing this model.
- 12. Line 141, please explain "the lapse rate tropopause".
- 13. Line 142, I assume Fig. 3 is produced from data for all three winters rather than Dec. 01, 2016.
- 14. Fig. 4, I assume both indices are computed over the North Atlantic (i.e., 60° W to 0° W and 35° N to 60° N)?
- 15. Fig. 5, If "both indices show a similar performance under different thresholds" is the only conclusion to be drawn from Fig. 5, I would suggest moving this 5-subplot figure into the appendix.
- 16. Fig. 6, "The contour lines represent the mixing ratio of O3 in ppmv of the respective QCTM simulation.". If so, which simulation did you use to plot the contour lines in subplot 6c?
- 17. Line 162, "significantly reduced" might be an over-statement.

- 18. Fig. 7b, is this the same as Fig. 6c?
- 19. Line 176, "changes of the gradient", what gradient are you referring to?